# Epidemiology and Prevention of Renal Cell Carcinoma

**DOI:** 10.3390/cancers14164059

**Published:** 2022-08-22

**Authors:** Tomoyuki Makino, Suguru Kadomoto, Kouji Izumi, Atsushi Mizokami

**Affiliations:** 1Department of Integrative Cancer Therapy and Urology, Kanazawa University Graduate School of Medical Science, 13-1 Takara-machi, Kanazawa 920-8640, Japan; 2Department of Urology, Ishikawa Prefectural Central Hospital, Kanazawa 920-8530, Japan

**Keywords:** renal cell carcinoma, epidemiology, prevention, intervening factors, hereditary diseases

## Abstract

**Simple Summary:**

The incidence of renal cell carcinoma (RCC) rises globally with the highest rates in developed countries. This indicates not only the impact of advanced imaging, but also an increased prevalence of modifiable risk factors such as smoking, obesity, and hypertension. This review will summarize the epidemiology and genetic variation of RCC, primary prevention-involving risk factors, and secondary prevention through early detection with a focus on biomarkers.

**Abstract:**

With 400,000 diagnosed and 180,000 deaths in 2020, renal cell carcinoma (RCC) accounts for 2.4% of all cancer diagnoses worldwide. The highest disease burden developed countries, primarily in Europe and North America. Incidence is projected to increase in the future as more countries shift to Western lifestyles. Risk factors for RCC include fixed factors such as gender, age, and hereditary diseases, as well as intervening factors such as smoking, obesity, hypertension, diabetes, diet and alcohol, and occupational exposure. Intervening factors in primary prevention, understanding of congenital risk factors and the establishment of early diagnostic tools are important for RCC. This review will discuss RCC epidemiology, risk factors, and biomarkers involved in reducing incidence and improving survival.

## 1. Introduction

Renal cell carcinoma (RCC) accounts for about 2.4% of all malignancies in adults, with more than 400,000 new cases diagnosed and about 180,000 deaths worldwide in 2020 according to GLOBOCAN data [1].

Increased detection of early RCC lesions in many patients is due to the widespread use of computed tomography (CT) and magnetic resonance imaging (MRI), and the 5-year survival rate for early RCC detection is as good as 93% [2,3]. However, there is a poor 5-year survival rate at 12% for RCC patients with metastases, and at least half of them require systemic drug therapy [4].

As it is generally resistant to cytotoxic chemotherapy and radiation therapy, RCC is classically considered an immunogenic tumor, and past immunotherapies have had some success, albeit inadequate [5]. Treatment of RCC has changed dramatically from 20 years ago, when the only options were surgical treatment and inadequate immunotherapy. A wealth of options are now available for RCC treatment options, including anti-vascular endothelial growth factor (VEGF) antibodies, VEGF receptor tyrosine kinase inhibitors, mammalian target of rapamycin (mTOR) pathway inhibitors, and immune checkpoint inhibitors (ICIs). Patient prognosis is improving [6].

However, improvement in the prognosis of advanced RCC is likely to be limited even with advances in pharmacotherapy. Therefore, understanding how to reduce the incidence of RCC and detect it early may help reduce the burden of disease, promote prompt treatment, and improve survival rates for this disease that claims 180,000 lives each year [1]. This presentation will provide an overview of the epidemiology and genetic variation of RCC, primary prevention-involving risk factors, and secondary prevention through early detection.

## 2. Epidemiology

RCC is the ninth most common malignant neoplasm in the United States (U.S.) and has been increasing in recent years [4]. Future lifestyle changes in Asia and Africa are expected to increase the number of RCC patients worldwide RCC as patients are most common in North America and Europe. This section describes the incidence and mortality of RCC.

### 2.1. Incidence

RCC is the 12th most common malignant neoplasm worldwide, excluding blood cancers, with approximately 431,000 new cases diagnosed, of which approximately 271,000 are in men and 160,000 are in women, based on 2020 estimates from GLOBOCAN data [1]. The age standardized rate (ASR) is 4.6 for the world average overall; 6.1 for men and 3.1 for women [1]. North America had the highest incidence at 12.2, followed by Australia and New Zealand at 10.2, and Europe at 9.5. ASRs in Asia and Africa were low at 2.8 and 1.8, respectively [1]. Those in Japan, Israel, and South Korea were high at 7.6, 7.5, and 6.5, respectively [1]. The high values in developed countries suggest that not only race but also lifestyle plays a role in the incidence.

### 2.2. Mortality

About 180,000 people died from RCC in 2020 accounting for 1.8% of all cancer patient deaths according to GLOBOCAN data, of which 116,000 were men and 64,000 were women [1]. The overall ASR was 1.8; 2.5 for men and 1.2 for women [1]. Comparing by region, mortality was higher in Eastern Europe and Latin America, with Slovakia at 4.7, Uruguay at 4.4, and Latvia at 4.3 [1]. The U.S. mortality rate in 2020 was 2.1, compared to a rate of 4.3/10,000 between 1992 and 1994, which may reflect improvements in diagnosis and treatment [7].

## 3. Risk Factors for RCC

Many risk factors for RCC have been reported, including genetic mutations. This section describes risk factors that cannot be changed.

### 3.1. Age, Gender and Race

Many sporadic RCCs are found in the elderly as it is generally a disease of the elderly. The average age of diagnosis in the U.S. is 64 years of age [7]. Most RCCs are derived from tubular tissue, with clear cell RCC (ccRCC), papillary cell RCC (pRCC), and chromophobe RCC (chRCC) accounting for more than 90% of the total [8]. The most common RCC is ccRCC (75%) of RCC and has a worse prognosis than the other two types [8]. Age at diagnosis of pRCC was significantly higher than ccRCC, but chRCC did not differ from ccRCC [9].

Most neoplasms are more common in males as shown in the GLOBOCAN data, and the same is true for RCC, with 1.5 times as many cases in males as in females [1]. This may be due to the fact that men have more lifestyle habits that contribute to carcinogenesis than women, as described below. Women have fewer pRCCs and more chRCCs in terms of histology, but the cause of this is unknown [9].

RCC incidence varies by ethnic group in the U.S., with a higher risk of RCC among Native Americans and African Americans, and lower among Asian Americans [3]. Factors contributing to this outcome may be due, not only to racial differences, but also to lifestyle factors such as diet and exercise, education, and access to medical care [10].

### 3.2. Hereditary Disease

Most RCC is sporadic, but some is due to specific germline gene mutations [11]. Therefore, hereditary diseases with those genetic mutations are risk factors, explaining their association with RCC (Table 1).

#### 3.2.1. VHL (Von Hippel–Lindau Disease)

Originating from tubular epithelial tissue and often metastasizing hematogenously to the lungs or bone, ccRCC is the most common type of RCC [12]. Von Hippel–Lindau (VHL) mutations gene on chromosome 3 are found in 90% of ccRCC [13]. In particular, VHL disease is the most well-known hereditary disease associated with RCC, accounts for 5% of ccRCC cases, and is a risk for bilateral RCC in young patients [14]. Decreased gene products associated with VHL mutations lead to increased expression of hypoxia-inducible factor (HIF)-1,2, a hypoxia-inducible factor that enhances angiogenesis into the tumor microenvironment and promotes tumor progression [15]. Increased HIF progresses tumors by involving growth factors such as transforming growth factor-β and platelet-derived growth factor, as well as glucose metabolism factors including glucose transporter [16].

The average age of RCC onset in patients with VHL disease is as young as 40–45 years, with a 70% lifetime risk, and the need for scheduled imaging studies for patients [17]. However, early surgical resection is not recommended after diagnosis because VHL-related RCC is often low-grade, and unlikely to metastasize if the size is less than 3 cm in diameter [18,19]. In addition, the mainstream practice is for nephron preservation as much as possible when performing surgery [20]. Recently, treatment with radiofrequency ablation is becoming more widespread [21].

BAP1 (Breast cancer susceptibility gene 1 associated protein 1 Tumor Predisposition Syndrome).

Breast cancer susceptibility gene 1 associated protein 1 (BAP1) tumor predisposition syndrome is an autosomal dominant syndrome associated with increased risk of malignant mesothelioma, uveal melanoma, cutaneous melanoma, basal cell carcinoma, and RCC [22]. BAP1 at 3p21.1 is involved in BRCA1-related tumor growth suppression, and germline mutations in patients with BAP1 tumor predisposition syndrome have been reported to be associated with familial RCC [23,24].

Although non-clear cell type has also been reported, BAP1-associated RCC subtype is mainly ccRCC [25]. Median age at diagnosis of BAP1-related RCC at 47–50 years is relatively young [22]. The risks associated with BAP1-related germline RCC are not well known because ccRCC with somatic BAP1 mutations are highly malignant, but early intervention and close follow-up may be necessary [26,27].

#### 3.2.2. MET (Mesenchymal Epithelial Transition Factor Receptor Hereditary Papillary RCC)

Cases of pRCC account for about 10–20% of all RCC cases, and most cases are solitary [28]. Two subtypes of pRCC, type 1 and type 2, with type 2 being more malignant [29].

Hereditary papillary RCC (HPRC) is an autosomal dominant inheritance syndrome and a risk for type 1 pRCC [28]. HPRC has not been reported to have non-renal involvement unlike other familial renal tumors [28]. The syndrome involves mutations in MET oncogene on chromosome 7, which promotes binding of the MET receptor to hepatocyte growth factor [30]. Activation of cell proliferation signaling and inhibition of apoptosis is triggered via the MET receptor and enhances cancer progression [30]. HPRC-related RCC is followed up to about 3 cm, and surgical therapy is recommended if the size is larger than that [31].

#### 3.2.3. FH (Hereditary Leiomyomatosis and RCC)

Hereditary leiomyomatosis and RCC (HLRCC) is an autosomal dominant hereditary cancer syndrome and a risk factor for pRCC type 2, uterine fibroids, and skin leiomyoma [32]. Patients with HLRCC have mutations in the gene for fumarate hydratase (FH), which catalyzes the hydration of fumaric acid to malic acid, located at 1q42-43 [33]. Inactivation of FH by mutation causes an increase in HIF-1, which promotes metastasis of malignant tumors through angiogenesis [34]. Fumarate accumulation associated with FH inactivation may induce epithelial-mesenchymal transition of tumor cells and promote metastasis [35]. The lifetime risk of renal cancer in HLRCC patients is approximately 15% for developing type 2 pRCC, thus patients require active surveillance and prompt resection of the tumor [36].

#### 3.2.4. SDHA/B/C/D (Hereditary Paraganglioma-Pheochromocytoma Syndrome)

Hereditary paraganglioma-pheochromocytoma syndrome (PGL/PCC) involves pathogenic variants of the succinate dehydrogenase (SDH) gene and is associated with the development of paragangliomas, pheochromocytomas, gastrointestinal stromal tumors, and RCC [37]. SDH is a mitochondrial enzyme complex composed of four protein subunits (SDHA/B/C/D), and involved in the Krebs cycle [38]. SDH-deficient RCC is a subtype of RCC characterized by loss of immunohistological expression of SDHB, which was added to the WHO tumor classification in 2016 [39]. Mutations have also been reported in other SDH genes (SDHA/C/D) and SDHB gene mutations in PGL/PCC patients cause RCC associated with disruption of the SDH complex [40]. Loss of SDHB causes disruption of the SDH complex, which is associated with 0.1–0.2% of all RCC [40].

Lifetime risk of RCC for SDHB mutation carriers is estimated to be about 5% [41]. Median age of onset of SDH-deficient RCC is as young as 40 years (14–76), with a wide range and multiple or bilateral occurrence in 30% of cases [42]. SDH-deficient RCC is highly malignant, with 33% developing metastases in one report [40]. Therefore, the presence of SDHB defects in the germ line may require adequate follow-up and early therapeutic intervention.

#### 3.2.5. TSC 1, 2 (TSC)

Tuberous Sclerosis Complex (TSC) is an autosomal dominant genetic disorder that can cause lesions in various sites, including the skin (angiofibroma), heart (rhabdomyoma), and kidney (angiomyolipoma (AML), cyst, RCC) [43]. TSCs loss-of-function germline mutations in either the tumor suppressor TSC1 or TSC2 genes [43]. The most common renal tumor in TSC is benign AML, of which ccRCC is the most common, with malignant AML and chRCC appearing in a few cases but malignant renal tumors are seen in about 5% of cases [44]. Because TSCs regulate the activity of mTOR complex 1 (mTORC1), inactivation of TSC1 or TSC2 results in tumor formation due to mTORC1 activation [43]. Therefore, the mTOR inhibitors sirolimus and everolimus are important drugs that reduce tumor size and suppress disease [45]. In patients with TSC, surgical resection is performed if RCC is suspected, and selective embolization is considered for AML if the size is 4 cm or larger [44].

#### 3.2.6. FLCN (Birt–Hogg–Dubé Syndrome)

Birt–Hogg–Dubé (BHD) is an autosomal dominant genetic disorder characterized by benign skin lesions, lung cysts with a high risk of pneumothorax, and multiple cysts in the liver and kidneys [46]. BHD is caused by a germline mutation of the FLCN gene on chromosome 17, resulting in bilateral multiple renal tumors in 12–34% of patients [46]. BHD-associated RCC is characterized by the formation of hybrid oncocytic tumors composed of chRCC and oncocytoma, with ccRCC and pRCC present in a minority of cases [47]. BHD-associated RCC is less malignant than VHL disease-associated ccRCC and HLRCC, but it can occur as young as in the 20 s, so active surveillance is needed [48]. Surgical resection with nephron preservation is recommended if the tumor is 3 cm in diameter [48].

#### 3.2.7. PTEN (PTEN Hamartoma Tumor Syndrome/Cowden Syndrome)

Phosphatase and tensin homolog (PTEN) hamartoma tumor syndrome/Cowden syndrome is an autosomal dominant disorder associated with germline mutations in the PTEN gene [49]. Located in chromosome 10, PTEN is a tumor suppressor gene involved in apoptosis, and cell cycle regulation via the mTOR pathway [50]. Therefore, PTEN mutations are associated with many tumorigenesis and predispose to multiple tumors including breast and colorectal cancer with an estimated 34% lifetime risk of RCC and various types of RCC, including ccRCC, pRCC, and chRCC [51]. HCC age of onset is young at around 40 years, and regular follow-up is necessary [51].

#### 3.2.8. CDC73 (Hyperparathyroidism-Jaw Tumor Syndrome)

Hyperparathyroidism-jaw tumor syndrome (HPT-JT) is an autosomal dominant genetic disorder that involves mutations in the cell division cycle (CDC73) gene [52]. The CDC73 gene is associated with parafibromin, which regulates cell proliferation and the cell cycle, and loss of normal parafibromin expression leads to decreased regulation of cell growth and tumorigenesis in certain tissues [52]. Characteristic tumors in HPT-JT patients include ossifying fibromas of the jawbone, parathyroid adenomas, uterine tumors, and renal tumors [52]. The most common renal tumors in HPT-JT patients are ccRCC and Wilms tumor [53]. Because HPT-JT patients may require multiple renal surgeries over their lifetime, nephron preservation is recommended in surgery [54].

**Table 1 cancers-14-04059-t001:** RCC-associated hereditary syndromes.

Syndrome	Gene	Gene Position	Protein	Histological Subtypes	Other Tumors	Pathways or Mechanism	Reference
VHL disease	VHL	3p25-26	pVHL	ccRCC	Hemangioblastoma (CNS) Cysts (kidneyl, pancreas) Pheochromocytoma (adrenal) Neuroendocrine tumor (pancreas) Cystadenoma (epididymis, ovary)	Overactivation of HIF-VEGF pathway	[13,14,15,16]
BAP1 tumor predisposition syndrome	BAP1	3p21	BRCA-associated protein	ccRCC	Breast cancer Melanoma (uveal, skin) Malignant mesothelioma Basal cell carcinoma Other multiple malignancies	Inactivation of cell cycle regulation and DNA damage repair	[22,23,24,25]
Hereditary papillary RCC	MET	7q31	HGFR	pRCC type1	NA	Overactivation of MET signalling pathway	[28,29,30]
Hereditary leiomyomatosis and renal cell carcinoma	FH	1q43	Fumarate hydratase	pRCC type 2	Leiomyoma (skin, uterus) Bladder cancer Breast cancer	Overactivation of HIF-VEGF pathway and EMT activation in tumor cells	[32,33,34,35]
Hereditary paraganglioma- pheochromocytoma syndrome	SDHA/B/C/D	5p15, 1p36, 1q21, 11q23	SDH	SDH-deficient RCC	Paraganglioma Pheochromocytoma GIST	Disruption of the tumor suppressor function of the SDH complex	[37,38,39,40]
Tuberous sclerosis complex	TSC 1, 2	9q34, 16p13	Hamartin, tuberin	Angiomyolipoma ccRCC	facial angiofibroma Intracardiac rhabdomyoma	Overactivation of mTORC1 signaling pathway	[43,44,45]
Birt-Hogg-Dubé syndrome	FLCN	17p11	Folliculin	Hybrid tmor chRCC ccRCC	Fibrofolliculoma Trichodiscoma	Overactivation of the mTOR pathway	[46,47,48]
PTEN hamartoma syndrome/Cowden syndrome	PTEN	10q23	PTEN	ccRCC pRCC chRCC	Breast cancer Endometrial cancer Thyroid cancer	Overactivation of the mTOR pathway	[49,50]
Hyperparathyroidism jaw tumour syndrome	CDC73	1q31	parafibromin	Hamartoma Wilms tumor	Parathyroid carcinoma Ossifying fibroma (jaw)	Inactivation of polymerase-related factor complex	[52,53]

VHL = Von Hippel-Lindau; RCC = renal cell carcinoma; ccRCC = clear cell renal cell carcinoma; CNS = central nervous system; HIF = hypoxia inducible factor; VEGF = vascular endothelial growth factor; BRCA = breast cancer susceptibility gene; BAP1 = BRCA1 associated protein 1; DNA = deoxyribonucleic acid; HGFR = hepatocyte growth factor receptor; pRCC = papillary RCC; NA = not applicable/available; FH = fumarate hydratase; EMT = epithelial mesenchymal transition; SDH = succinate dehydrogenase; GIST = gastrointestinal stromal tumor; TSC = tuberous sclerosis complex; mTOR = mammalian target of rapamycin; mTORC1 = mTOR complex 1; chRCC = chromophobe RCC; PTEN = phosphatase and tensine homolog; CDC73 = cell division cycle 73.

### 3.3. Drugs and Chemicals

Over-the-counter analgesics are frequently used worldwide and have been reported to be associated with RCC risk.

An aspirin study found an increased risk in non-US studies (five studies, pooled relative risk [RR]: 1.17; 95% CI: 1.04–1.33), but no difference overall (pooled RR: 1.10; 95% CI: 0.95–1.28) [55]. However, use of acetaminophen (pooled RR: 1.28; 95% CI: 1.15–1.44) and other NSAIDs (pooled RR: 1.25; 95% CI: 1.06–1.46) was associated with increased risk of RCC [55].

Substances that have been associated with specific occupational exposures are not associated with renal cancer, such as bladder cancer and malignant mesothelioma. However, trichloroethylene has strong degreasing power, which is used as an organic solvent, and has been used in the past for cleaning, specifically semiconductor cleaning. It has been reported to be associated with kidney cancer. This substance is highly carcinogenic and is a predisposing risk factor for RCC, along with liver cancer and lymphoma [56].

## 4. Primary Prevention

The comprehensive VITamin And Lifestyle (VITAL) study in the U.S. reported that lifestyle disease such as obesity and hypertension, and smoking increased the risk of developing RCC [57]. Moreover, European reports also suggested that high obesity, hypertension, and hyperglycemia in men, and high body mass index (BMI) in women increase the risk of developing RCC [58]. This section outlines primary prevention in terms of dietary habits and smoking, which are believed to increase the risk of developing RCC, as well as risk factors such as obesity, hypertension, and diabetes, which are typical of lifestyle-related diseases.

### 4.1. Tobacco Smoking

Tobacco smoking has been linked to many common cancers including RCC. Tobacco smoke contains a mixture of carcinogens implicated in the etiology of RCC [59]. Epidemiologic evidence for a causal relationship with tobacco includes a dose-response relationship between risk and the amount smoked per day, and a decrease in risk with longer years of smoking cessation [60]. In the U.S., the VITAL study revealed that a 22.5 pack-year smokers had a more than 50% increased RCC risk compared to nonsmokers [57]. Meta-analysis of more than 24 articles has recently shown that a pooled RR for RCC incidence is 1.31 (95% confidence interval [CI], 1.22–1.40) for all smokers, 1.36 (95% CI, 1.19–1.56) for current smokers, and 1.16 (95% CI, 1.08–1.25) for former smokers [61]. A recent report from the Japan Public Health Center-based Prospective Study (JPHC study) revealed that heavy smokers (≥40 pack-years) had an increased risk of RCC (hazard ratio [HR] 1.50; 95% CI, 1.01–2.25), albeit only when data for men and women were combined, although the effect of smoking on RCC incidence has not been well investigated in Asian populations [62].

### 4.2. Alcohol Consumption

Moderate alcohol consumption has been reported to have a protective effect on RCC incidence compared to abstinence [59]. A very large European prospective observational study examined the association between alcohol consumption and the risk of RCC within the European Prospective Investigation into Cancer and Nutrition (EPIC) [63]. The odds ratio (OR) of cancer risk was 0.72 (95% CI, 0.54–0.96) in the group consuming 12–24 g of alcohol per day, and 0.67 (95% CI, 0.50–0.89) in the group consuming 24–60 g of alcohol per day compared to the non-drinking group [63]. This suggests that alcohol consumption may have a prophylactic effect in the development of RCC. The VITAL study in the U.S. conversely found no association between alcohol intake and RCC [57]. There was a slight, but not significant inverse association between alcohol consumption and RCC incidence in the JPHC study [62]. Although a recent meta-analysis confirmed that alcohol consumption from wine, beer, and liquor is associated with a decreased risk of RCC. Statistically significant inverse associations were found for wine (RR 0.82; 95% CI, 0.73–0.91) in women, and for beer and liquor (RR 0.87; 95% CI, 0.83–0.91 and RR 0.95; 95% CI, 0.92–0.99, respectively) in men when these associations were examined by gender [64].

### 4.3. Eating Habits

In terms of diet, the group with a median red meat intake of 62.7 g per 1000 kcal had an increased risk of cancer with a HR of 1.19 (95% CI, 1.01–1.40) compared with the group with a median intake of 9.8 g [65]. This correlation was limited to women in another prospective study, with an OR of 2.03 (95% CI, 1.14–3.63) for cancer risk in the group with a daily red meat intake of >80 g compared to the group with a daily intake of <10 g [66].

Consumption of fruits and vegetables (especially cruciferous vegetables) has been reported to be associated with reduced risk of RCC [67,68]. However, no significant association was found between fruit or vegetable intake and RCC risk in the EPIC study (HR 1.03; 95% CI, 0.97–1.08 and HR 0.97; 95% CI, 0.85–1.11, respectively) despite a wide range of intake [69], as well as in the VITAL study (HR 1.02; 95% CI, 0.71–1.46 and HR 0.76; 95% CI, 0.52–1.11, respectively) [57].

The risk of median daily sodium intake (3.4 g in men and 2.8 g in women) was presumably mediated by the development of hypertension [70], although this risk of was associated with a HR of 1.40 (95% CI, 0.99–1.97) for risk of cancer compared with 1.9 g in men and 1.5 g in women.

On the other hand, there is growing evidence that coffee can prevent several chronic diseases, including cancer and metabolic diseases [71,72,73] although coffee is one of the most widely consumed beverages worldwide. Cafestol and kahweol are natural diterpenes extracted from coffee beans and have particularly been shown to have a wide variety of bioactive properties, including anti-inflammation, anti-angiogenesis, and anti-tumorigenic activity [74]. Recent in vitro experimental results have revealed that anti-proliferative and anti-migratory properties of kahweol acetate and cafestol in human renal cancer cells [75], and anti-angiogenic properties of cafestol palmitate and kahweol palmitate in human microvascular endothelial cells [76]. Hence, coffee–containing compounds may have an anticancer aspect.

### 4.4. Maintaining an Appropriate Body Weight

The VITAL study confirmed that obesity is significantly associated with RCC risk (BMI ≥ 35 vs. < 25 kg/m^2^; HR 1.71; 95% CI, 1.06–2.79) [66]. The RR estimate corresponding to approximately 5 kg of body weight increases the risk of RCC by 25% for men and 35% for women [57]. Another study has been reported that not only the elderly, but also young males have an increased risk of cancer with a HR of 2.43 (95% CI, 1.54–3.83) in the group with a BMI over 27.5 kg/m^2^ compared to the counterpart under 22.5 kg/m^2^ [77]. Men with a BMI ≥ 27 kg/m^2^ had a HR of 1.99 (95% CI, 1.04–3.81) for cancer risk in the JPHC study, compared with a BMI of 23–24.9 kg/m^2^. In contrast, overweight women were positively associated with RCC risk, albeit not statistically significant (BMI ≥ 25 vs. 21–24 kg/m^2^; HR 1.55; 95% CI, 0.76–3.18). Interestingly, an increased risk of cancer was also observed in thin men with a BMI < 21 kg/m^2^ (HR 1.86; 95% CI, 1.01–3.45), indicating a U-shaped association not seen in other countries [78].

### 4.5. Comorbidities

#### 4.5.1. Hypertension

Hypertension was independently associated with RCC risk (HR 1.70; 95% CI, 1.30–2.22) [57] and evidence has been reported that hypertension predisposes to RCC. A recent meta-analysis in the VITAL study of 18 prospective studies further supports a positive association between hypertension and RCC risk. A history of hypertension was associated with 67% increased risk of RCC, and that for every 10 mmHg increase in blood pressure, the risk of RCC increased by 10–22% [79]. The biological mechanism underlying the relationship between hypertension and RCC remains unclear, but is hypothesized to involve chronic renal hypoxia and lipid peroxidation due to the formation of reactive oxygen species [80,81]. In contrast, the efficacy of the antihypertensive agents such as renin-angiotensin system (RAS) inhibitors against RCC metastasis have been reported by a number of basic and meta-analytic studies [82]. Hypertensive patients taking RAS inhibitors are expected to be potential chemopreventive effect on RCC due to smaller tumor size and fewer incidence of metastasis [83], and have a significantly prolonged overall survival after surgical treatment of RCC [84,85].

#### 4.5.2. Diabetes

Type 2 diabetes has been associated with an increased risk of several types of cancer [86,87], but its relationship to RCC is still unclear. Indeed, no relationship between diabetes and RCC was observed in the VITAL study [57], while in the Nurses’ Health study, type 2 diabetes was significantly associated with an increased risk of RCC in women (HR 1.60; 95% CI, 1.19–2.17) [88]. High levels of glucose among men were additionally associated with increased risk of RCC (HR 3.75; 95% CI, 1.46–9.68) [58]. Moreover, a meta-analysis of nine cohort studies indicates a positive association between diabetes and risk of RCC (RR 1.42; 95% CI, 1.06–1.91), stronger in women (RR 1.70; 95% CI, 1.47–1.97) than in men (RR 1.26; 95% CI, 1.06–1.49).

## 5. Secondary Prevention

Screening programs improve survival rates by detecting and treating RCC in its early, curable stages. The ideal screening modalities have not yet been determined and there are currently no diagnostic modalities for the early detection of RCC other than incidental radiological discovery although it has been postulated that screening for RCC may be a cost-effective strategy through downstaging the disease, reducing the prevalence of metastatic tumors and associated expenditure relating to systemic therapies. Some candidate molecules in blood and urine have been reported as biomarkers for RCC recently by searching serum microRNAs and urinary proteins or by metabolomics, proteomics, and amino acid profile analysis. There are no results or sufficient evaluations of screening in a large general population, and they have not reached a practical stage.

### 5.1. Serum and Urine Biomarkers as a Screening Modality

As an alternative to radiologic screening, the use of tumor markers with high sensitivity and specificity is desirable, but no readily available and clinically validated biomarkers for RCC screening are available. However, several urine and serum biomarkers have been proposed as potential screening tools. The collection of this biological fluid can also be attractive due to its quick, easy, and non-invasive method in clinical practice. The most promising urine biomarkers are aquaporin 1 (AQP1) and perilipin 2 (PLIN2). These concentrations are sensitive and specific biomarkers for the early non-invasive detection of clear cell or papillary subtypes of RCC [89,90]. The sensitivity and specificity of both biomarkers were 85–92% and 87–100%, respectively. This had an area under the curve (AUC) of 0.95 and 0.91 for AQP1 and PLIN2, respectively. Moreover, biomarkers can distinguish RCC from healthy controls, benign renal tumors, and non-renal urological cancers [91]. In addition, it is well known to reflect many types of kidney damage and has limited specificity as a diagnostic biomarker for RCC although urinary kidney injury molecule-1 is also significantly higher in patients with RCC than controls and has diagnostic sensitivity for RCC [92,93]. In contrast, a combined three-marker assay based on nicotinamide N-methyltransferase, L-plastin, and non-metastatic cell 1 protein was a promising novel serum marker. They were developed with a sensitivity of 95.7%, specificity of 90%. The diagnostic accuracy of AUC is 0.932 for RCC versus healthy controls. However, this assay is limited in its ability to distinguish between RCC and benign renal tumors [94]. Recently, epigenetic phenomena can regulate gene expression, particularly DNA methylation, microRNAs (miRNAs), and long non-coding RNAs (lncRNAs), and such changes have been reported to be frequently associated with various clinical subgroups of RCC [95,96,97]. Incidentally, miRNAs were the most widely studied in terms of potential non-invasive biomarkers for RCC, while a limited number of studies focused on DNA methylation. Due to the high specificity and diagnostic potential of lncRNAs, further efforts should be made for the wider investigation of these novel biomarkers in the future [98].

#### 5.1.1. DNA Methylation

Alterations in DNA methylation occur early during cancer development and are observed even in the precancerous state in the case of clear cell RCC (ccRCC) [99,100], with an increased promoter hypermethylation frequency in higher stage and grade tumors [101]. The most innovative study on non-invasive RCC detection was reported by Nuzzo et al. [102], who used cell-free methylated DNA immunoprecipitation and high throughput sequencing (cf-MeDIP-seq) for sensitive detection of early-stage tumors. The researchers identified differentially methylated regions (DMRs) between patients and control groups to construct a classifier and performed cf-MeDIP-seq on cell-free DNA samples of plasma. The top 300 DMRs were able to accurately detect all stages of RCC with an AUC of 0.99 and 0.86 in plasma and urine samples, respectively. Furthermore, the plasma sample strongly discriminated between RCC and urothelial bladder cancer with an AUC of 0.98. However, due to its complexity, the clinical application of such a classification method is currently limited.

#### 5.1.2. MiRNAs

Emerging evidence has suggested that miRNAs may be promising novel biomarkers for the diagnosis of RCC [103,104]. The pooled sensitivity and specificity of miRNAs for the diagnosis of RCC were 0.85 (95 % CI, 0.77–0.90) and 0.84 (95 % CI, 0.70–0.92), respectively according to a meta-analysis conducted to comprehensively evaluate the diagnostic potential of miRNAs in RCC. The value of AUC was 0.91 (95 % CI, 0.88–0.93), suggesting that the diagnostic accuracy of miRNAs achieved a relatively high level [105]. Another report found a new panel of 5-miRNAs (including miR-193a-3p, miR-362, miR-572, miR-28-5p, and miR-378) that can clearly distinguish RCC patients from non-cancer controls, demonstrating the value of the 5-miRNA panel as a clinical diagnostic aid to detect early-stage RCC [106] (Table 2). Another study confirmed high serum miR-122-5p and miR-206 levels in ccRCC patients compared to healthy controls. Further, higher miR-122-5p and miR-206 levels were directly associated with pathological tumor-stage, -grade, and metastatic RCC, and with a shorter progression-free survival and overall survival, indicating potential prognostic noninvasive biomarkers [107]. The miR-210 is the most widely studied circulating miRNA in the case of RCC [108,109,110], and the upregulation of miR-210 combined with two other miRNAs, miR-155 [109] and miR-1233 [111] were found. All studies found that circulating miR-210 levels in RCC patients were increased compared to healthy controls despite widely differing experimental conditions. It is well known that miR-210 is expressed in response to hypoxia, mainly through HIF-1α, a key player of renal carcinogenesis [112]. However, it is worth mentioning that upregulation of circulating miR-210 was also found in various other malignancies [113], and further validation with appropriate controls seems to be mandatory. Urine is an attractive new promising source considering different body fluids for miRNA biomarker detection. Notably, the value of miR-210-3p has not only been demonstrated to be significantly elevated in urine samples collected from two independent cohorts of ccRCC patients at the time of surgery compared to samples from healthy donors [114,115], but also resulted in a significant decrease in response to treatment [116]. Besides miR-210-3p, recent studies have observed significantly higher levels of miR-122, miR-1271, and miR-15b in ccRCC urine specimens compared to controls [117]. Moreover, direct association was obtained between RCC size and miR-15a expression values in urine, and miR-15a expression differentiated RCC from benign renal tumors with a specificity of 98.1% and sensitivity of 100%, with an AUC of 0.955 [118]. Further, as shown by Pinho et al., the level of promoter methylation of miR-30a-5p is elevated in ccRCC and metastatic urine samples compared to healthy donors and non-metastatic ccRCC, suggesting that miR-30a-5p^me^ levels might be an indicator of disease progression and metastatic processes [119]. Cancer-suppressive miRNAs have also been identified contrary to the aberrant expression of oncogenic miRNAs in urine samples. Song et al. showed lower levels of urinary exosomal miR-30c-5p in ccRCC patients compared to controls [120].

**Table 2 cancers-14-04059-t002:** microRNAs as potential diagnostic biomarkers in RCC.

miRNA	Sample	Significant Expression of miRNA	AUC (95% CI)	Sensitivity (%)	Specificity (%)	Reference
miR-193a-3p miR-362 miR-572 miR-28-5p miR-378	Serum	Significantly higher level of miR-193a-3p, miR-362, and miR-572 whereas markedly lower level of miR-28-5p and miR-378 in RCC patients. 5-miRNA panel showed a high level in the stage I RCC compared with HCs.	Panel of 5 miRNAs: 0.801 (0.731–0.871)	80	71	[106]
miR-122-5p miR-206	Serum	High level of miR-122-5p and miR-206 in metastasized and in advanced pT-stage ccRCC. Association of these miRNAs with progression-free and overall survival.	miR-206: 0.733 (0.616–0.849)	57.1	83.8	[107]
miR-210	Serum	Overexpression of serum miR-210 level in RCC patients compared to HCs.	miR-210: 0.77 (0.65–0.89)	65	83	[108]
miR-210 miR-155	Serum	High level of serum miR-210 and miR-155 in ccRCC patients compared to HCs.	miR-210: 0.87 (0.79–0.95)	82.5	80	[109]
miR-210 miR-1233	Serum	High level of serum miR-210 and miR-1233 in ccRCC patients compared to HCs.	miR-210: 0.69 (0.61–0.77) miR-1233: 0.82 (0.75–0.89)	miR-210: 70 miR-1233: 81	miR-210: 62.2 miR-1233: 76	[111]
miR-210-3p	Urine	Upregulation of miR-210-3p in ccRCC tissues and in urine samples. Correlation between urinary levels of miR-210-3p and response to treatment.	NA	NA	NA	[112]
miR-122 miR-1271 miR-15b	Urine	Significantly higher level of miR-122, miR-1271, and miR-15b in the ccRCC urine specimens compared to HCs.	Combination of 3 miRNAs: 0.96 (0.88–1.04)	100	86	[117]
miR-15a	Urine	The expression of miR-15a differentiated RCC from benign renal tumors.	0.955	98.1	100	[118]
miR-30a-5p^me^	Urine	Significantly higher miR-30a-5p^me^ level in urine from ccRCC patients compared to HCs. Higher miR-30a-5p^me^ levels independently predicted metastatic dissemination and survival.	0.684 (0.584–0.784)	83	53	[119]
miR-30c-5p	Urine	Low expression of miR-30c-5p in ccRCC patients compared to HCs.	0.819 (0.739–0.899)	68.6	100	[120]

ccRCC = clear cell renal cell carcinoma; HC = healthy control; AUC = area under the curve; CI = confidence interval; NA = not applicable/available.

#### 5.1.3. lncRNAs

Recently, lncRNAs have been shown to contribute to the development of almost all cancer types, including RCC [121,122]. lncRNAs have been implicated in many processes related to cancer development and progression, including cell cycle regulation, proliferation, apoptosis, senescence, migration, invasion, and drug resistance as well as miRNAs [98]. PVT1, LET, PANDAR, PTENP1, HOTAIR, NBAT1, LINC00963, KCNQ1OT1, GAS5, CADM-AS1, RCCRT1, MEG3, SPRY4-IT1, HIF1A-AS, MALAT1, and others [123,124,125] are the lncRNAs implicated in RCC. It has been reported that five significantly down-regulated lncRNAs including LET, PVT1, PANDAR, PTENP1, and LINC00963 in ccRCC patients when compared to healthy controls with an AUC of 5-lncRNAs panel equal to 0.90 and 0.82 for the training and testing sets of samples, respectively [126] (Table 3). Moreover, the panel also significantly distinguished ccRCC from benign renal tumors. The serum levels of GIHCG [127] and LINC00887 [128] were found to be significantly elevated in RCC patients compared to healthy controls, and could be identified with a sensitivity of 87.0% and specificity of 84.8%, with an AUC of 0.920, and with a sensitivity of 67.1% and specificity of 89.9%, with an AUC of 0.803, respectively. In addition, although GIHCG and LINC00887 had a higher expression, both signatures interestingly have a background related to RCC cell proliferation and migration in vitro [127,128], and were related with advanced clinical tumor-stage and shorter survival. Thus, the specificity of lncRNAs for RCC seems to be equal to or greater than that of miRNAs and DNA methylation, and lncRNAs may have the potential to be promising and well-performing novel RCC biomarkers. The lncRNAs have remarkable potential as not only diagnostic markers but also as therapeutic targets; high expression of MALAT1 has been suggested to be a biomarker for early detection of lymph node metastasis or a predictor of reduced survival in RCC patients [129,130]. In addition, poor prognosis in RCC patients [122] is associated with TCL6, NBAT-1, SPRY4-IT1, RCCRT1, GAS5, and CADM1-AS1.

**Table 3 cancers-14-04059-t003:** lncRNAs as potential diagnostic biomarkers in RCC.

lncRNA	Sample	Significant Expression of lncRNA	AUC (95%CI)	Sensitivity (%)	Specificity (%)	Reference
LET PVT1 PANDAR PTENP1 LINC00963	Serum	A risk model of serum 5-lncRNA signature could distinguish benign renal tumors from ccRCC samples.	Panel of 5 lncRNA: 0.900/0.823	79.2/67.6	88.9/91.4	[126]
GIHCG	Serum	Serum GIHCG accurately discriminated between RCC patients and HCs, as well as between early stage RCC patients and HCs. Positively correlation of increased GIHCG expression with advanced clinical stage, Fuhrman grade, and poor prognosis.	0.920 (0.866–0.974)	87.0	84.8	[127]
LINC00887	Serum	Upregulated LINC00887 in tumor tissues and serum of RCC patients compared to HCs. Relationship between high expression of LINC00887 and shorter overall survival.	0.803 (0.735–0.872)	67.1	89.9	[128]
MALAT-1	Tissue	Elevated MALAT-1 levels significantly correlated with decreased overall survival in RCC patients (hazard ratio, 2.97; 95% CI, 1.68–5.28)	NA	NA	NA	[129]

ccRCC = clear cell renal cell carcinoma; HC = healthy control; AUC = area under the curve; CI = confidence interval; NA = not applicable/available.

### 5.2. Metabolomics

Previous studies have identified metabolites of ccRCC using tissue samples evaluated with the Global Metabolomics protocol [131]. Pathways associated with these metabolites were found to be relevant for diagnosis and prediction of malignancy including glutathione, tryptophan, and glycolysis. Additionally, the glycoglycerolipid, carnitine, and tocopherol pathways are potentially diagnostic, while the tricarboxylic acid cycle, nucleotide sugars, and inositol pathways are associated with malignancy [131]. Urine is theoretically an ideal tool for the study of urinary tract diseases because low molecular weight compounds (e.g., low molecular weight metabolites) can be freely filtered out. Therefore, urine metabolomics may be useful for metabolic profiling and biomarker discovery for urological cancers [132,133]. Recently, untargeted metabolomic analysis of urine samples from RCC patients and healthy subjects revealed that p-cresol glucuronide can be a diagnostic marker for RCC [134]. Furthermore, analysis of urine samples from RCC patients one year after nephrectomy revealed that isobutyryl-_L_-carnitine and _L_-proline betaine could be prognostic markers [134]. Others have reported the following [135]: five metabolites (_L_-glutamic acid, lactate, _D_-sedoheptulose 7-phosphate, 2-hydroxyglutarate, and myoinositol) in a diagnostic predictive model for ccRCC and four metabolites (_L_-kynurenine, _L_-glutamine, fructose 6-phosphate, and butyrylcarnitine) in a predictive model for clinical stage III/IV ccRCC were extracted. The sensitivity and specificity of the diagnostic prediction model were 93.1% and 95.0%, respectively, with an AUC of 0.966. Sensitivity and specificity of the clinical stage prediction model were 88.5% and 75.4%, respectively, with an AUC of 0.837 [135]. Guida et al., studied the first comprehensive metabolomics analysis of incident RCC to be conducted using a prospective design, using pre-diagnostic blood samples from up to 1305 RCC case–control pairs from five prospective cohort studies [136]. In this study, 25 metabolites were found to be robustly associated with risk after 1416 metabolites associated with the development of RCC were analyzed. The relationship between lipid metabolites and future RCC risk could theoretically capture the increased uptake of lipid metabolites due to preclinical cancer development although the majority of metabolites were classified as glycerophospholipids.

### 5.3. Proteomics

Proteomics studies of RCC using a variety of mass spectrometry (MS) techniques have been applied to many different types of samples, including tumor cell lines, serum, tissue, and urine [137]. A high molecular weight extracellular matrix protein that plays an important role in cellular attachment and cell spread [138], the plasma levels of fibronectin 1 (FN1), are significantly elevated in localized and metastatic RCC patients compared to a control group [139,140]. Another study suggested that FN1 mRNA expression is higher in RCC compared to normal renal tissue and correlates with advanced disease, suggesting that FN1 mRNA expression might serve as a marker for RCC aggressiveness [141]. Moreover, state-of-the-art label-free quantitative proteomics method evaluated the RCC proteome in comparison to non-cancer renal tissue. This has identified 596 proteins including von Willebrand factor, Ectonucleotide pyrophosphatase/phosphodiesterase family member 3, adipose differentiation-related protein (ADFP), Coronin 1A, thymidine phosphorylase, nicotinamide N-methyltransferase, fatty-acid binding protein 5, annexin A4, laminin, vimentin, NADH dehydrogenase, metallothionein, ubiquitin carboxyl-terminal hydrolase isozyme L1, and L-xylulose reductase [142] as the important cancer-associated proteins highlighted. In particular, ADFP has been suggested as a potential biomarker for diagnosis, prognosis prediction, and treatment efficacy [89]. A recent detailed quantitative proteomics identified 10,160 unique proteins, 955 of which were significantly regulated between tumor and normal adjacent tissue [143]. As a result, PLOD2, FERMT3, SPARC, and SIRPα, were identified as the four candidate secreted biomarkers that are highly expressed proteins unaffected by intra- and inter-tumor heterogeneity [143]. In addition, SPARC showed a marked increase in urine samples from ccRCC patients and is a promising marker for disease detection in body fluids.

### 5.4. Lipidomics

Lipidomics is an independent and emerging field within metabolomics [144], and lipid metabolism dysfunction has been found to be associated with the pathogenesis of many diseases, including ovarian cancer [145], prostate cancer [146], and breast cancer [147]. Significant alterations in linoleic acid metabolism have also been observed in many other cancer types (colorectal, bladder, and RCC) associated with inflammatory disorders, immune responses, and cell proliferation [148]. Global lipid profiling analysis by MALDI-FT-ICR MS identified 39 lipids that most discriminate between tumor and healthy tissue or tumor recurrence and non-recurrence status [149]. Elsewhere, a lipidomic approach identified lipids that differed in presence between RCC tissue and normal cortex in the same affected area. Higher levels of ether-type phospholipids, cholesterol esters, and triacylglycerols, as well as by lower levels of phospholipids (except phosphatidylcholines) and polyunsaturated fatty acids [150] distinguished the cancerous tissues. Comprehensive liquid chromatography-MS utilizing plasma metabolomics and lipidomics was used to globally identify the plasma profiles of 64 bladder cancer patients, 74 RCC patients, and 141 healthy controls [148]. An apparent separation was observed between cancer (bladder cancer and RCC) plasma samples and controls, with an AUC by plasma metabolomics and lipidomics of 0.985 and 0.993, respectively. In addition, homocysteine thiolactone, acetylcysteine, methionine sulfoximine, 16-Hydroxy-10-oxohexadecanoic acid, 9S,10R-Epoxy-6Z-nonadecene, (10E,12Z)-(9S)-9-Hydroperoxyoctadeca-10,12-dienoic acid, avenoleic acid, and 9,10,13-TriHOME were the eight metabolites found which had good predictive ability for discriminating between bladder cancer, RCC, and controls;. Furthermore, a recent discovery-based lipid profiling study of human serum samples reported not only a 16-lipid panel discriminating ccRCC patients from controls with 77.1% accuracy in an independent test set, but also a 26-lipid panel discriminating between early and late ccRCC with 82.1% accuracy in an independent test set [151].

### 5.5. Amino Acid Profile Analysis

Extensive research has been conducted to determine whether serum amino acid levels can be a biomarker for RCC. One study reported that the concentrations of 26 amino acids in serum collected preoperatively from 189 RCC patients and 104 age- and sex-matched controls showed statistically significant changes in the concentrations of 15 amino acids, with a decrease in 13 and an increase in two amino acids. Furthermore, the value of AUC was 0.81, based on a logistic regression model using eight amino acids (cysteine, ornithine, histidine, leucine, tyrosine, proline, valine, and lysine). This same model also had predictive value in terms of overall survival and tumor recurrence in patients with RCC, suggesting that serum amino acid levels may be useful as a screening tool in identifying patients with RCC and predicting outcome [152]. Another study showed that logistic regression models can be constructed using serum levels of histidine, glutamine, 1-methylhistidine, and norvaline, and are particularly valuable for predicting patients with early clinical tumor-stage, low Fuhrman grade ccRCC, and for assessing prognosis in ccRCC patients [153]. The logistic regression model score based on serum levels of these amino acids was also reported to be an independent prognostic predictor of progression-free survival in patients with ccRCC [153]. Moreover, significant differences in plasma/serum free amino acid profiles were observed between ccRCC patients and healthy controls [154]. In particular, serum histidine and plasma tryptophan were able to correctly classify 85.5% of control and 84.7% of case samples with the logistic regression model [154].

## 6. Conclusions

The incidence and mortality of RCC vary widely around the world. Hereditary risk factors cannot be modified, but behavioral and environmental factors and comorbidities can be improved. Prevention efforts targeting smoking, obesity, hypertension, diabetes, and occupational exposure are therefore important.

Secondary prevention, mainly early detection, is also important in addition to prevention of outbreaks. Adequate follow-up for hereditary diseases and early detection of sporadic cases, which account for the majority of cases, may reduce the number of deaths due to RCC.

A significant number of biomarkers for RCC have been studied and reported for diagnostic and prognostic purposes. However, none are yet accurate enough to be widely used in clinical practice. Therefore, further search and elucidation of new markers for RCC and research on existing markers should be continued.

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
