# Peer review of "Epidemiology and Prevention of Renal Cell Carcinoma"

_cancers, 2022, doi:10.3390/cancers14164059_

Round 1

Reviewer 1 Report

The paper is suitable for publication in the present form.

The manuscript is well written, comprehensive, and well balanced. All sections on it are updated. I have enjoyed very much the manuscript, and the paragraphs speaking about primary and secondary prevention deserve in my opinion a special mention. I believe the manuscript should be accepted as it is.

Author Response

Answer: Thank you very much for your positive comments.

Reviewer 2 Report

This article provides a comprehensive review of the literature on the epidemiology, risk factors, and secondary prevention of renal cell carcinoma (RCC). I would suggest the following to improve this manuscript: 

1. Overall, the English used is readable and no major grammar corrections are needed. Yet, the text would benefit from occasional rephrasing or revisions to the wording used. 

2. An example of the above is when the authors state "Most RCCs are sporadic, but some involve specific genetic mutations" It would be more accurate and clear to say "specific germline genetic mutations" 

3. Also, the authors refer to some of the paper sections as "chapter", although this manuscript is not a book.

4. The authors should insert a specific reference for the "GLOBOCAN" data cited in some parts of the review. 

5. The section on Mortality only cites general global and specific US data, but not data from non-US countries or regions. I recommend adding information on mortality rates from RCC in specific non-US countries (as was included in the section on incidence of RCC)

6. The authors should add a definition of "lncRNAs" abbreviation used at least once in the manuscript. 

Author Response

  1. Overall, the English used is readable and no major grammar corrections are needed. Yet, the text would benefit from occasional rephrasing or revisions to the wording used. 
  2. An example of the above is when the authors state "Most RCCs are sporadic, but some involve specific genetic mutations" It would be more accurate and clear to say "specific germlinegenetic mutations" 

Answer: Thank you very much for your suggestions. We have corrected some of the wording, including the part you pointed out. (Page3 line1)

  1. Also, the authors refer to some of the paper sections as "chapter", although this manuscript is not a book.

Answer: Thank you very much for your comments. Chapter" was used in two places, which has been corrected.

  1. The authors should insert a specific reference for the "GLOBOCAN" data cited in some parts of the review. 

Answer: Thank you very much for your suggestions. References were inserted for GLOBOCAN data.

  1. The section on Mortality only cites general global and specific US data, but not data from non-US countries or regions. I recommend adding information on mortality rates from RCC in specific non-US countries (as was included in the section on incidence of RCC)

Answer: Thank you very much for your comments. We have additionally mentioned mortality rates in other regions. (Page2 line19-21)

  1. The authors should add a definition of "lncRNAs" abbreviation used at least once in the manuscript. 

Answer: Thank you very much for your comments. lncRNA is an abbreviation for long non-coding RNAs. We have listed them in the first place of appearance. (Page9 line37)